# The Transition to a Renewable Energy Electric Grid in the Caribbean Island Nation of Antigua and Barbuda

Patrick Hoody [1], Andrew Chiasson [1,2] and Robert J. Brecha [2,3,*]

[1] Department of Mechanical and Aerospace Engineering, University of Dayton, Dayton, OH 45469, USA; hoodyp1@udayton.edu (P.H.); achiasson1@udayton.edu (A.C.)
[2] Renewable and Clean Energy Program, University of Dayton, Dayton, OH 45469, USA
[3] Hanley Sustainability Institute, University of Dayton, Dayton, OH 45469, USA
* Correspondence: rbrecha1@udayton.edu

**Abstract:** The present study describes the development and application of a model of the national electricity system for the Caribbean dual-island nation of Antigua and Barbuda to investigate the cost-optimal mix of solar photovoltaics (PVs), wind, and, in the most novel contribution, concentrating solar power (CSP). These technologies, together with battery and hydrogen energy storage, can enable the aim of achieving 100% renewable electricity and zero carbon emissions. The motivation for this study was that while most nations in the Caribbean rely largely on diesel fuel or heavy fuel oil for grid electricity generation, many countries have renewable resources beyond wind and solar energy. Antigua and Barbuda generates 93% of its electricity from diesel-fueled generators and has set the target of becoming a net-zero nation by 2040, as well as having 86% renewable energy generation in the electricity sector by 2030, but the nation has no hydroelectric or geothermal resources. Thus, this study aims to demonstrate that CSP is a renewable energy technology that can help assist Antigua and Barbuda in its transition to a renewable energy electric grid while also decreasing electricity generation costs. The modeled, optimal mix of renewable energy technologies presented here was found for Antigua and Barbuda by assessing the levelized cost of electricity (LCOE) for systems comprising various combinations of energy technologies and storage. Other factors were also considered, such as land use and job creation. It was found that 100% renewable electricity systems are viable and significantly less costly than current power systems and that there is no single defined pathway towards a 100% renewable energy grid, but several options are available.

**Keywords:** renewable energy; concentrating solar power; hydrogen storage; just transition; electric grid; Caribbean

## 1. Introduction

Antigua and Barbuda is a small dual-island nation in the Caribbean, the most north-eastern island of the Lesser Antilles [1]. Of the total population, 97% is on Antigua, although the islands are comparable in land area, with the island of Antigua having an area of 281 km$^2$ and the island of Barbuda having an area of 161 km$^2$ [2]. The tropical climate has very little variation throughout the year, with the median temperature in any month not falling below 25 °C, based on measurements from the past 30 years [3]; Antigua receives around 2782 h of sunshine a year [4]. The key environmental issues for Antigua and Barbuda include water management; the minimal freshwater resources on the islands and the impacts of deforestation from colonial sugar plantations allow rainfall to run off more quickly [5]. Susceptibility to tropical storms and hurricanes further exacerbates these environmental issues, leading to increasing efforts toward resilience and adaptation to climate change. One potential solution the nation has looked into for water purification is using wind power for desalination, which would require a significant additional amount of electricity [6] but would increase resilience.

Electricity generation in Antigua and Barbuda is nearly completely reliant on imported petroleum products. Diesel energy comprises 89% of the 87.45 MW of installed capacity for the nation [7]. The electricity production and distribution are operated by two companies: Antigua Power Company (APC) and Antigua Public Utilities Authorities (APUA) [8]. APC is a private company that owns the generating capacity, whereas APUA is the utility company that distributes the electricity and charges for its consumption. The companies work together very closely, as APC sells the electricity it produces to APUA to then sell to its customers. These plants, combined with the other small backup generators that are owned by individuals and businesses, contribute to the nation emitting just under 200,000 metric tons of carbon dioxide per year from the electricity sector and 650,000 metric tons in total from the energy sector. The nation is hoping to reduce that number drastically and has set a goal of reducing emissions in the energy sector [7].

Antigua and Barbuda's commitments to the Paris Agreement, as outlined in their NDC, include targets of becoming a net-zero nation by 2040 and having 86% renewable energy generation in the electricity sector by 2030. The additional targets to be achieved by 2030, as identified explicitly in the NDC, include having 100 MW of renewable energy capacity for the grid, a target of constructing 20 MW of wind energy, 50 MW of renewable energy capacity owned by farmers who can sell to others, and 100 MW of renewable energy capacity, owned by social investment entities such as non-governmental organizations, bus associations, or any other businesses registered as social investors. The NDC also identifies the need to establish a framework to achieve these goals by 2024. Finally, a specific goal relevant to the current work is that 30,000 homes, or 50% of pre-2020 homes, should have backup renewable energy systems for at least 4–6 h of energy. The solar resource for Antigua is approximately 6 kWh/m$^2$/day, and therefore, solar PV is a well-suited technology for this goal. In 2020, the residential sector in Antigua consumed 103 GWh of energy [7]. With the current total household count at 30,213, this implies an average of 3400 kWh of energy consumed per household [9].

However, with increasing electricity consumption in homes and the introduction of electric vehicles (EVs), which the nation aims to progress towards, it is likely that annual household consumption will increase. An average, personally owned vehicle would require just over 1750 kWh in a year [10,11]. When combined, a household would require around 5250 kWh per year. In Antigua, a solar panel can produce upwards of 1500 kWh/kW$_{peak}$ [12]. Thus, a 4 kW array of solar panels will produce about 6000 kWh in a year. This will be larger than the required 5250 kWh households from past data and EV introduction, which will give room for the electrification of homes. With the NDC target of 30,000 homes having solar PV systems, approximately 120 MW of rooftop solar PVs will have to be installed to achieve this NDC target.

Recent legislation and policies, including the National Energy Policy and Environmental Protection and Management Act [13], set out goals of reducing carbon emissions from the energy sector by 62% by 2027 and 90% by 2030 [13]. These and other policies cover the plans of protecting their local environments, implementing renewable energy into their energy system, ensuring affordable, equitable, and accessible energy to all, and developing standards for buildings and vehicles to increase their energy efficiency.

In order to achieve the goals of transitioning to renewable energy and reducing carbon emissions in the energy sector, it is necessary to understand which technologies would best fit the nation's renewable resources. With solar energy being a viable and abundant resource, both solar photovoltaics (solar PVs) and concentrated solar power (CSP) are considered in this work, along with wind energy, which is also a part of the NDC targets [14]. These three sources, along with energy storage technologies (batteries and hydrogen), will be the most viable low-carbon and market-ready options for power generation in the country based on the nation's renewable resources. CSP is a technology that has not been considered for the region, but we postulate that it is well-suited based on its operational needs of a high number of solar hours and direct solar incidence.

Geothermal energy and hydropower are two technologies that neighboring islands can potentially utilize, given the abundance of geothermal resources in parts of the Caribbean. However, Antigua and Barbuda (together with Barbados) do not have geothermal energy as an electricity-generating option [15]. Hydropower is also not an available resource in Antigua and Barbuda, although some nearby nations have been able to take advantage of that resource.

## 2. Methodology

### 2.1. Electric Grid Simulation Model Description

PyPSA (Python power system analysis) is an open-source Python framework used to model energy systems [16]. In the case of Antigua, we have used PyPSA to investigate the cost-optimal mix of solar PV, wind, and CSP, together with energy storage, with the aim of achieving 100% renewable electricity and zero emissions [14] in the timeframe of 2035–2040. In PyPSA, the system is configured with several components: buses, loads, generators, links, and stores. Each bus has generators (plants that produce electricity) and loads connected to it, as well as energy storage (stores); links connect buses to one another. There are four buses in this model, with one containing the load and all the generation technologies except CSP (Antigua), a separate bus, and Antigua CSP for all components of the CSP system, as well as a bus for each utility battery charge and discharge and for hydrogen generation and storage. A schematic of the process flow is shown in Figure 1.

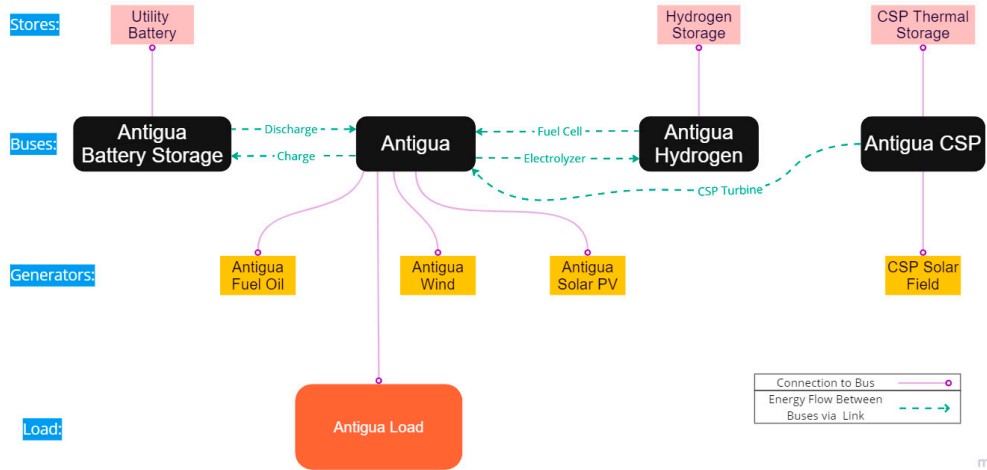

**Figure 1.** PyPSA flow diagram showing the main components of the modeled system. "Buses" represent the main centers of connectivity and are separated into a main system bus and three storage buses, each with a "Stores" technology. The "Generators" are generating technologies that are connected to the buses. The "Load" represents the hourly demand for electricity during the year. "Links" represent the efficiencies in terms of storing energy and/or the methods by which the energy is transferred between "Buses".

For the Antigua bus, there is a load, Antigua Load, which contains the hourly electricity demand data for the country, with three generators: diesel power generation (Antigua Fuel Oil), wind-power (Antigua Wind), and solar PV (Antigua PV). Wind and solar PV can charge the Antigua battery storage or generate hydrogen if they produce more energy than the load in a given hourly interval since they are all linked to the Antigua bus. The Antigua CSP bus has a generator for the concentrating solar power (CSP) solar field and can charge the CSP Thermal Storage store since they are both under the same bus. Finally, any of the stores can meet the demand of the Antigua Load. When the load is larger than all the incoming power, the stores are able to meet the remaining loads. The links can take into account capital costs if used to represent, for example, an electrolyzer, fuel cell, or a turbine converting thermal energy into electricity in the CSP module. The process efficiencies associated with the charging and discharging of the batteries and the conversion

of electricity to hydrogen (electrolyzer) and then hydrogen back to electricity (fuel cell) are assigned to the links. Since there is no load for the Antigua CSP bus, the energy generated by the CSP solar field will either try to meet the Antigua Load through the link, CSP Turbine, or charge the CSP Thermal storage, with the associated capital cost for the turbine and thermal storage used as the input for optimization.

This model energy system presented here for Antigua can be easily modified to analyze other areas in the Caribbean or other regions around the world by using the appropriately determined renewable energy resources, capital, and marginal costs, along with the country's hourly electricity demand profile.

## 2.2. Load Description

The load data used in the model are based on the estimates of typical loads for Antigua, but hourly load data (as needed for the model) are not publicly available. Thus, a load model was used to represent the approximate load pattern for every hour of the year in Antigua, based on actual open-source hourly data for Martinique [17], which have been scaled to Antigua based on the total and peak demands that are publicly available [15]. Although the two countries differ greatly in population, the load patterns are similar. In order to gain an estimation of the demand profile for the year 2035, the model was multiplied by a factor of 1.2 to account for both a growing power demand of approximately 10% based on past trends and an estimated additional 10% load based on increased vehicle electrification [18]. Although based on [18], a slightly lower value for the penetration of electric vehicles (EVs) of 35% by 2035 has been assumed for this model. This constant factor assumes that the load profile remains the same in the future, an assumption that may not hold under higher penetrations of EVs and grid interactivity, which are also included as targets in their NDC [14].

## 2.3. Generator Descriptions

The current electricity production system in Antigua is heavily reliant on heavy fuel oil generators. In order to address the NDC targets, we constructed several model scenarios that do not allow for any fossil fuels, which implies large amounts of solar and wind energy, together with storage. For every scenario, hourly generation was needed to determine how the variable renewable energy sources, wind, solar PV, and CSP, would contribute to satisfying the demand for each hour of the year.

CSP is a technology that has not been used in the Caribbean, but Antigua and the Caribbean receive a lot of direct sunlight. Spain has many CSP plants in Alvarado, Majadas, and Orellana [19], which share similar climates, getting a similar amount of total sun hours in a year as Antigua [20]. Thus, it is logical to test if this technology will decrease the costs of energy for Antigua. This technology also has a very long storage time in some plants, such as the DEWA IV CSP-PV hybrid plant in Dubai, which has 10 h of storage for its trough CSP, so it can hold energy that can be distributed for the late peak demand that occurs most days in Antigua [21]. Utility batteries today are essentially held to 6 h of storage at best but are estimated to reach up to around 10 h of storage in the future [22]. However, CSP thermal storage has reached up to 14 h in some plants, and research is being carried out to extend these times as well [23]. This combination of reasons motivates including CSP in this analysis and is specifically relevant for Antigua, where there are no other dispatchable renewable energy resources to complement wind and solar PV.

The hourly output of each generator was needed for PyPSA modeling. The solar PV and solar CSP used hourly output for the systems using the NASA POWER database, which has hourly data for various weather conditions from several years for Antigua [24]. A more detailed explanation of the hourly CSP output calculations can be found in Appendix D. Hourly wind energy generation estimates were taken from re-analysis data for Antigua [12]. As an input into PyPSA, each source is scaled to unity and can then be used to calculate the system output depending on the capacity installed. Worksheets are used to determine

CSP system properties and are included in the GitHub repository referenced at the end of the manuscript.

### 2.4. Stores Descriptions

PyPSA has two types of storage methods available: (i) storage units that are based on the fixed hours of storage and variable power, and (ii) stores with a fixed output power and a variable number of hours of storage for each unit. The latter was used for this present study. The capital costs of the technologies in units of USD/MWh were provided as inputs. The amount of energy in the storage units in the first hour of running the model was set to be equal to the energy stored in the last hour of running the model for optimization and to avoid the end effects of storage either having to be initially charged or to avoid seeking a solution in which all stored energy was discharged at the end of the modeling period. The model was constructed for an annual period, allowing annual cycling in a multi-year simulation.

Both utility-scale batteries and hydrogen were used as storage technologies in this study, in addition to thermal storage from CSP capacity. Utility-scale batteries were only given a limitation in two of the groupings of scenarios conducted in this analysis. That limit was set to 6 h of storage for the scenario where all the renewable energy technologies considered in this study were included, and 10 h was given for the sensitivity tests. This type of storage is compatible with wind and solar PV energy. The CSP thermal storage was given a limit of 24 h for all the scenarios to reflect dual 12 h storage tanks. Dual storage tanks are used in many CSP plants around the world [25]. The storage systems that are found in the results are effectively given not in terms of thermal storage but rather in terms of the resulting electrical output. The thermal storage would be around three times as large as the results shown due to the generation efficiency needed to convert thermal input to electrical output. This difference intrinsically takes into account the capital costs because the storage and generation capital costs defined in Tables 1 and 2 are based on electrical output and electrical storage. Additionally, thermal storage has a non-zero minimum that would need to be maintained, but for the purpose of this study, having a minimum of zero is suitable for the model. The charge and discharge efficiencies of the batteries were 95% (which results in 90.25% overall efficiency) [26], with efficiencies being implemented through the PyPSA links.

For energy storage and electricity production with hydrogen, the model included electrolysis, hydrogen storage, and fuel cells for converting hydrogen back into electricity. The efficiency of electricity to hydrogen through electrolysis is set at 75%, and the fuel cell turning the hydrogen back to electricity has an efficiency of 60% [27]. The electrolysis and fuel cell technologies will be characterized through the links connected to and from the hydrogen store, which takes into consideration each of these efficiencies and capital costs. Since hydrogen can be stored (essentially) indefinitely, the size of the storage can be much larger. The hydrogen storage was not given any limitations for that reason.

The CSP plant generates thermal energy that is then converted into useful electrical energy before reaching the grid. However, since the power generated from the "CSP Solar Field" generator has already taken this into account, the storage will be accounted for in terms of useful energy rather than in terms of thermal energy. All energy that is generated or stored in the CSP bus will be transferred to the load under the Antigua bus via the CSP Turbine link, which has an associated capital cost per MW of capacity.

### 2.5. Technology Economics

The following tables outline the capital costs of each type of technology. There is also a marginal operating cost, mainly fuel, associated with the diesel generator, which is included in Table 1. The costs for all technologies are estimates for the period of interest, in about 2035, and are, therefore, necessarily very approximate. However, we consider these estimates to be conservative, as wind and PV systems have already achieved these cost levels in larger countries today. The capital costs for the stores are shown in Table 2.

**Table 1.** Generator and link capital costs.

| Generator | Capital Cost (USD/kW) | Marginal Cost (USD/MWh) |
|---|---|---|
| Diesel [28] | 1800 | 170 |
| Wind [29] | 1350 | N/A |
| Solar PV [30] | 880 | N/A |
| Concentrating Solar Power Solar Field [31] | 2640 | N/A |
| Concentrating Solar Power Turbine [31] | 760 | N/A |
| Hydrogen Electrolyzer (Link) [27] | 1000 | N/A |
| Hydrogen Fuel Cell (Link) [27] | 500 | N/A |

**Table 2.** Store capital costs.

| Stores | Capital Cost (USD/kWh) |
|---|---|
| Utility Battery [22] | 143 |
| CSP Thermal Storage [31] | 50 |
| Hydrogen Storage [27] | 33.33 |

To interpret the results given from PyPSA, a levelized cost of electricity (LCOE) was calculated by dividing the total system capital costs, plus marginal costs, by the total yearly demand in MWh. Since the model is creating an energy system based on only one year of data, the capital costs of all technologies need to be modified to represent a real levelized cost of electricity. Thus, the capital cost values above were multiplied by a capital recovery factor (CRF) using a discount rate of 7%. Further details are described in Appendix B.

### 2.6. Limits on Technologies

A set of constraints were implemented in the model corresponding to minimum capacities in (renewable) technologies already installed, as well as maximum capacities as estimated from physical and technological limits. For example, wind energy was set to maximum values in increments of 25 MW to understand how the solar PV, CSP, and storage technologies would respond to the extra demand the wind was not covering. Although there is a published estimate of a potential of 400 MW of wind power capacity in Antigua and Barbuda [15], a wind energy capacity of larger than about 50 MW appears unlikely, given concerns about the tourism industry. Solar PV capacity was not given a limit as the results proved to be reasonable. With the inclusion of rooftop solar PV and the knowledge that the nation has plentiful solar resources, it was logical not to put a limit on the technology. The published estimates for a solar potential of 27 MW [7] appear to be far too low, given that the NDC targets are set well above this potential and according to the physical land-use reasons described below. The modeling and results in this study are reflective of the NDC targets [14]. Government-identified land for renewable energy has been identified, including the land near the Parham Ridge Wind farm and the existing solar PV Bethesda array, which equate to around 0.8 km$^2$ of land [18]. Between the approximate 90 km$^2$ of agricultural land on the island, parking areas, commercial buildings, etc., land areas of up to 4 km$^2$ could be utilized for the remaining land needs for solar PVs [32]. Storage limits were based on the realistic limitations of the current technologies.

### 2.7. Land Use Description

Antigua's total area of 281 km$^2$ is also a limited resource to be considered, and consequently, for each of the scenarios, the total land area requirement was estimated. In this analysis, only solar PV and CSP were taken into consideration. The direct land use of wind turbines is small, although the effective use of land and the visual footprint can be quite large. The land area used per MW was determined to be 17,000 m$^2$/MW (1.7 hectares/MW) [33] for solar PV and 26,000 m$^2$/MW (2.6 ha/MW) for CSP [23].

*2.8. Job Creation and Destruction*

Important to the energy system transition is the concept of a "just transition" from the current paradigm. Although the just transition involves many aspects, such as societal engagement and democratic processes, one key piece is that of changes in employment [34]. Employment impact in the electricity sector in Antigua and Barbuda was calculated for each scenario using multipliers that were assigned to the different technologies based on construction and installation, operation and maintenance, and fuel supply [35]. There is also a regional factor that considers the current workforce capabilities of the nation. As Antigua and Barbuda has minimal solar PV and no other renewable sources installed, there will be a lack of experience in the installation and maintenance within the workforce to begin this transition. Then, as time goes on and the workers gain experience, the number of people needed for the same amount of work will decrease. Appendix C details all the factors that contribute to each of these defined categories. These factors are defined in jobs/MW for operation and maintenance for the overall capacity of each technology or job-years/MW for construction and installation for the capacity added each year for each technology. For each scenario, a logistic growth to the final capacities, as given in terms of optimization, was used to estimate the yearly job additions for each technology, as well as the jobs needed for operation and maintenance [36].

**3. Results**

*3.1. Current Electricity System*

Scenarios were developed with the assumption of finding electricity system configurations that are consistent with the NDC goals of 86% renewable energy generation for electricity and 30,000 homes with solar systems [14]. To compare the different scenarios on an equivalent basis, a model of the current electricity system was created as a baseline. In all cases, a "green field" approach was used, with the assumption that no current system is in place. Therefore, the total cost and levelized costs shown in our results will be based on annualized accounting with all capital and marginal costs accounted for. Thus, the scenario that represents a theoretical optimum corresponding most closely to the current system in Antigua, including fuel oil generators, is likely to be counterfactual because it would assume the replacement of the current capacities a decade in the future; the assumption is that new generators would have to be bought by then, even if the country was to continue down a business-as-usual pathway. In the case of fuel oil generators, the marginal cost is taken as USD 0.17/kWh and is a significant fraction of the total annualized cost.

Table 3 gives the LCOE found by the model when constrained to the technologies currently in use, as well as the corresponding LCOE in the year 2035 with the assumed increase in demand by 20% and assuming that no more solar will be installed in that time. As would be expected, the LCOE will remain the same; these results are useful as a baseline to compare with other scenarios and tell us what the relative costs would be for maintaining or replacing the current system.

**Table 3.** Current and 2035 diesel-based grids.

| Antigua Current and Future Business-as-Usual System | | |
|---|---|---|
| Scenarios | Current System | 2035 System |
| LCOE (USD/MWh) | 189 | 190 |
| Diesel (MW) | 53.5 | 66.8 |
| Solar PV (MW) | 9 | 9 |

*3.2. Wind, Solar PV, and Batteries Scenarios*

In the first set of scenarios, only solar PV, wind, and batteries are included. These technologies are the most common renewable energy sources that are widely used today and are explicitly included in Antigua and Barbuda's NDC goals. Thus, it was important to understand if these technologies alone would create a reliable and economical system.

As described in the Methodology, the costs of the systems are based on the capital cost estimates for 2035. Table 4 shows the results of three scenarios with these technologies alone, differing in the maximum amount of wind capacity allowed as part of the solution space. These already represent 100% renewable energy, zero-carbon-emission scenarios and demonstrate a slightly lower LCOE than the current and "business-as-usual" systems.

**Table 4.** Renewable technologies scenarios with wind, solar PV and batteries only.

| | Solar PV, Wind, and Batteries Only | | |
|---|---|---|---|
| **Scenarios** | **25 MW Wind Max** | **50 MW Wind Max** | **100 MW Wind Max** |
| LCOE (USD/MWh) | 169 | 155 | 154 |
| Land Use ($km^2$) | 10.3 | 7.6 | 7.1 |
| Solar PV (MW) | 608 | 448 | 420 |
| Wind (MW) | 25 | 50 | 73 |
| Battery Storage (MWh) | 1588 | 1796 | 1725 |
| Hours of Storage (Hours) | 26.5 | 29.9 | 28.75 |

These scenarios were arranged in a way that increased the wind capacity from 25 MW to 100 MW. All these scenarios are at or below the LCOE of the baseline system by about 15%. However, the storage times were not limited in this scenario, which allows an estimate of the kinds of storage needed to bridge periods of low wind and solar resources that may occur during some hours of the year. As seen in Table 4, storage times of up to nearly 30 h are required for each of the scenarios. An assumed 70 MW was given for the maximum output of the batteries, given the system peak load of 67.7 MW. These scenarios are not very reliable solutions for an energy system based on battery storage, which might be limited to 8–12 h, even in the future, and even with newer technologies, such as redox-flow batteries [37]. It should also be noted that wind power plays an important role, and when limited to lower capacities, as is the case in the first two scenarios, optimization requires the use of as much wind capacity as possible.

### 3.3. Introduction of CSP with Wind, Solar PV, and Batteries

One of the main contributions of this present work was to examine the utility of CSP as an option in a (near) 100% electricity system. Although levelized cost reductions were seen for the solar PV, wind, and long-term storage system, it is important to find more realistic storage options.

CSP was included in three further scenarios with the other renewable technologies that were used in the previous scenarios. Again, upper limits were set to the wind capacity; however, for these scenarios, there were also limits set on each of the energy storage systems. CSP thermal storage was given a limit of 24 h with no limit on the battery storage. Results for this scenario configuration are shown in Table 5.

The model results of these scenarios show significant cost reductions of 28–33% when compared to the current system LCOE. This decrease was due to the large energy storage provided by the CSP thermal tanks. Thus, the required utility-battery storage was only 8 h, which is still not a storage time that is commonly available today but is projected for batteries in the near future. These results show a similar trend: increasing the wind energy only somewhat affects the system LCOE. As in the first set of scenarios, limiting the wind capacity to 50 MW or less leads to a solution that drives toward the maximum of that constraint, but about 50 MW of wind capacity is the unconstrained optimum. Additionally, the land use for each of these scenarios was significantly less than those in the previous scenarios.



**Table 5.** Renewables with CSP scenarios.

| | Renewables with CSP | | |
|---|---|---|---|
| **Scenarios** | **25 MW Wind Max** | **50 MW Wind Max** | **100 MW Wind Max** |
| LCOE (USD/MWh) | 136 | 128 | 127 |
| Land Use (km$^2$) | 7.5 | 6.6 | 6.5 |
| Solar PV (MW) | 380 | 370 | 367 |
| Wind (MW) | 25 | 50 | 53 |
| CSP Solar Field (MW) | 40 | 13 | 12 |
| CSP Turbine (MW) | 60 | 60 | 60 |
| Battery Storage (MWh)/Hours of Storage | 541/7.75 h | 565/8 h | 568/8 h |
| CSP Thermal Storage (MWh)/Hours of Storage | 1440/24 h | 1440/24 h | 1440/24 h |

### 3.4. Addition of Hydrogen Storage

The final additional technology considered was that of electrolysis-generated hydrogen that could be stored for long periods of time if necessary and then be converted into electricity using fuel cells [38]. Table 6 shows the results for a selection of system configurations with hydrogen as an option.

**Table 6.** All technologies scenarios.

| | Renewables with CSP and Hydrogen Generation and Storage | | |
|---|---|---|---|
| **Scenarios** | **No Restrictions** | **Limited Wind (<25 MW); CSP Must be Included** | **Limited Wind (<25 MW); No CSP** |
| LCOE (USD/MWh) | 122 | 130 | 127 |
| Land Use (km$^2$) | 5.1 | 6.5 | 7.3 |
| Solar PV (MW) | 289 | 378 | 432 |
| Wind (MW) | 58 | 25 | 25 |
| CSP Solar Field (MW) | 8 | 4 | N/A |
| CSP Turbine (MW) | 53 | 60 | N/A |
| Battery Storage (MWh)/Hours of Storage | 420/6 h | 420/6 h | 420/6 h |
| CSP Thermal Storage (MWh)/Hours of Storage | 1280/24.2 h | 1440/24 h | N/A |
| Hydrogen Storage (MWh) | 1721 | 885 | 2000 |
| Hydrogen Electrolyzer (MW) | 23 | 41 | 67 |
| Hydrogen Fuel Cell (MW) | 15 | 17 | 45 |

These scenarios yield similar system LCOEs to those without hydrogen. However, it shows that the CSP could be replaced completely with hydrogen generation and storage to produce a slightly lower system LCOE. By limiting wind to 25 MW or less, the optimal system would exclude CSP. To test the sensitivity with respect to these two options, one scenario was run to identify the LCOE with a limited wind system that must include CSP to understand the cost of the system with both CSP and hydrogen storage. Although the scenario that requires CSP to be included has a higher LCOE than the scenario without, the uncertainty in technology costs in 2035 implies that these two cases are practically indistinguishable in cost. Including CSP decreases land use by nearly 1 square kilometer, which is a potential advantage. The combined systems with all technologies in use find that the storage times for utility batteries and CSP thermal storage are storage times that are in place today, although they are possibly slightly less common.

### 3.5. Small Diesel Systems Remaining

The scenarios presented thus far are considerably cheaper than the current system. In all cases, however, the inclusion of diesel has not been incorporated into these hypothetical electricity system models. Antigua is seeking to obtain 86% of its energy production from renewable energy by 2030, so including some diesel generation in the system could be a very useful way to help meet this goal and to help, in general, with the transition to 100% renewable energy generation. The results from the scenarios with all the technologies included, including diesel generation, either unrestricted or limited to maximum capacities of 10 and 5 MW, are shown in Table 7. Figures 2 and 3 illustrate for these latter two cases the relative infrequency of use of diesel generation during the year and how in these scenarios, diesel generators play a small but important role in ensuring that demand is met in all hours.

**Table 7.** Scenarios with small amounts of diesel generation.

| | Small Diesel Contribution | | |
|---|---|---|---|
| **Scenarios** | **No Restrictions** | **10 MW Diesel** | **5 MW Diesel** |
| LCOE (USD/MWh) | 83 | 106 | 119 |
| Land Use (km$^2$) | 1.8 | 4.3 | 5.7 |
| Solar PV (MW) | 107 | 239 | 306 |
| Wind (MW) | 90 | 51 | 57 |
| CSP Solar Field (MW) | 0 | 10 | 19 |
| CSP Turbine (MW) | 0 | 60 | 60 |
| Diesel (MW) | 38.5 | 10 | 5 |
| Battery Storage (MWh)/Hours of Storage | 195/2.8 h | 420/6 h | 420/6 h |
| CSP Thermal Storage (MWh)/Hours of Storage | 0 | 1433/23.9 h | 1440/24 h |
| Renewable Energy Penetration (%) | 88 | 97 | 99 |

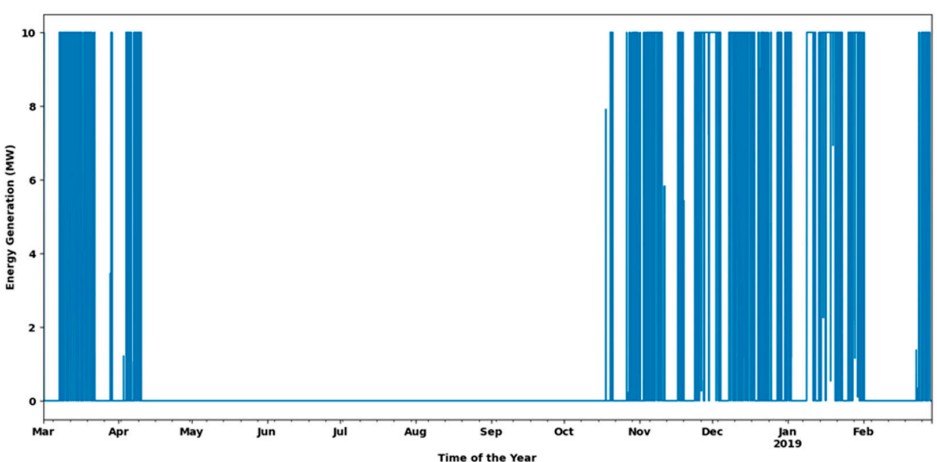

**Figure 2.** Diesel generation: 10 MW.

The results shown in Table 7 have lower LCOEs than the previous sets of scenarios, although the 5 and 3 MW diesel systems are roughly the same as not having any diesel capacity. These diesel systems do not actually generate much electricity, with capacity factors of between 16% for the largest diesel system and only 7% for the 5 MW system. Thus, diesel capacity acts as a flexible resource that runs only a few hundred hours each year, and it is implicitly assumed to be able to power on and off at will, a characteristic that may not reflect the reality of the larger diesel generators that are typical for Caribbean islands. However, there are technologies that allow for ramping to be carried out within a minute, thus allowing these scenarios to become a reality with the larger plants needed at low capacity factors [39]. It should be noted that the scenarios with larger amounts of diesel generation would require significantly less land than any of the other scenarios shown. In all these scenarios, the renewable energy generation is higher than the NDC target of 86%

by 2030. Thus, any of these systems would allow Antigua to reach its goal. However, this could be accomplished with diesel generation that is constant, as their diesel plants would operate now, albeit at a low-capacity factor.

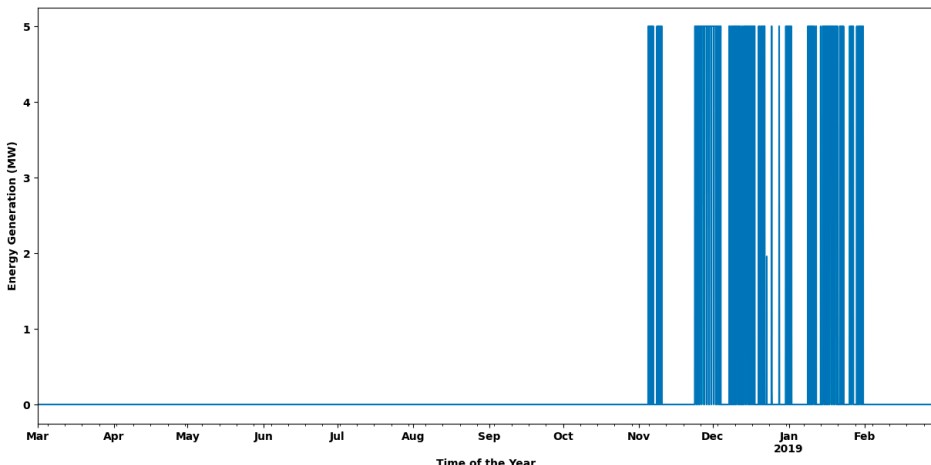

**Figure 3.** Diesel generation: 5 MW.

### 3.6. Summary of Scenarios

Based on the assumed load for 2035, a constant 7 MW of diesel generation could be used to accomplish 86% of the energy generated by renewable energy. The results in Table 8 show various scenarios that could be utilized in the transition to 100% renewable energy generation. These results show that the LCOE will be competitive with the other completely renewable and small-scale diesel generators that can be ramped on and off quickly and easily. However, this model shows, again, that having no CSP or hydrogen storage will drive up the LCOE quite significantly. Thus, at least one of these technologies could be extremely beneficial to the system to help decrease LCOE, even if the technologies do not play the most important role in the system. It should be noted that the results of numerous scenario variants (found on the GitHub repository) indicate that the optimal solutions for nearly every scenario result in a CSP system with a 60 MW turbine with 24 h of storage (1440 MWh).

**Table 8.** Constant diesel generation: 7 MW.

| Constant 7 MW Diesel Contribution | | | |
|---|---|---|---|
| **Scenarios** | **No Restrictions** | **No Hydrogen** | **No Hydrogen or CSP** |
| LCOE (USD/MWh) | 126 | 131 | 147 |
| Land Use (km$^2$) | 4.8 | 5.0 | 6.4 |
| Solar PV (MW) | 273 | 274 | 379 |
| Wind (MW) | 34 | 38 | 28 |
| CSP Solar Field (MW) | 7 | 14 | N/A |
| CSP Turbine (MW) | 35.5 | 60 | N/A |
| Utility Storage (MWh)/Hours of Storage | 473/6.8 h | 476/6.8 h | 1318/18.8 h |
| CSP Thermal Storage (MWh)/Hours of Storage | 846/23.8 h | 1413/23.55 h | N/A |
| Hydrogen Storage (MWh) | 1123 | N/A | N/A |
| Hydrogen Electrolyzer (MW) | 7 | N/A | N/A |
| Hydrogen Fuel Cell (MW) | 8 | N/A | N/A |

### 3.7. Sensitivity Tests

A sensitivity analysis regarding the capital cost of solar PV and CSP and the marginal cost of diesel generation per MWh was carried out. The scenarios were run with an assumed solar PV capital cost of USD 1200/kW and USD 550/kW rather than a baseline of USD 880/kW. The CSP capital costs were increased by a factor of 1.2 (USD 3168/kW for solar field, USD 912/kW for turbine, and USD 60/kWh for storage) and decreased by a factor of 0.8 (USD 2112/kW for solar field, USD 608/kW for turbine, and USD 40/kWh for storage) as sensitivity checks. Finally, the baseline marginal cost of diesel generation, USD 0.60/liter [40], was increased to USD 1/liter as a check for the sensitivity of the scenarios to higher fossil fuel costs.

For a system that had no restrictions for solar PV, wind, CSP, and utility storage and a higher cost of fuel oil, a system LCOE of USD 127/MWh was found but had no fossil fuel capacity, as shown in Table 9. That is, a higher fuel oil cost results in the system without fossil fuels being the lowest-cost optimal solution.

**Table 9.** Diesel fuel and solar PV sensitivity tests.

| Sensitivity Tests | | | |
| --- | --- | --- | --- |
| Scenarios | Diesel Fuel Increased | Solar PV CAPEX Increased | Solar PV CAPEX Decreased |
| LCOE (USD/MWh) | 127 | 153 | 108 |
| Land Use (km$^2$) | 6.6 | 7.3 | 8.2 |
| Solar PV (MW) | 367 | 341 | 461 |
| Wind (MW) | 52 | 25 | 25 |
| CSP Solar Field (MW) | 12 | 57 | 15 |
| CSP Turbine (MW) | 60 | 60 | 60 |
| Diesel (MW) | 0 | N/A | N/A |
| Utility Storage (MWh)/Hours of Storage | 568/8 h | 519/7.4 h | 577/8.25 h |
| CSP Thermal Storage (MWh)/Hours of Storage | 1440/24 h | 1440/24 h | 1440/24 h |

For the sensitivity tests regarding the capital costs for solar PV and CSP, the following constraints were implemented: 25 MW for wind energy, a maximum battery storage of 10 h, and a maximum CSP storage time of 24 h. The results are shown in Tables 9 and 10.

**Table 10.** CSP sensitivity tests.

| Sensitivity Tests Continued | | | |
| --- | --- | --- | --- |
| Scenarios | CSP CAPEX Increased | CSP CAPEX Decreased | CSP Extended Storage Time |
| LCOE (USD/MWh) | 143 | 127 | 124 |
| Land Use (km$^2$) | 7.9 | 7.4 | 6.6 |
| Solar PV (MW) | 435 | 359 | 367 |
| Wind (MW) | 25 | 25 | 53 |
| CSP Solar Field (MW) | 19 | 49 | 12 |
| CSP Turbine | 60 | 60 | 40 |
| Utility Storage (MWh)/Hours of Storage | 606/8.7 h | 529/7.75 h | 568/8 h |
| CSP Thermal Storage (MWh)/Hours of Storage | 1440/24 h | 1440/24 h | 1440/36 h |

The cost of solar PV impacts the system LCOE the most, with less sensitivity in the resulting CSP storage time capacity and cost. Given the strong tendency in the past for solar PV costs to decrease and for the cost of oil on global markets to fluctuate, both the main results and the sensitivity tests indicate that a system based mainly on solar PVs and with a gradual phase-out of fossil fuel generation will be economically advantageous.

### 3.8. Land Use Results

Land use is a key factor in these systems, as renewable energy technologies take up considerably larger areas than fossil fuel plants do. Many of the scenarios required up to 8 km$^2$ of land for the solar PV and CSP plants. Based on the results from the scenarios, the land use of these technologies can be calculated. The land use for solar PV and CSP are both relevant to the available land in Antigua and the NDC targets that are set by the nation.

As indicators for land use, there are two large areas currently in use by the energy sector, the West Indies Oil Company facility and the site of the current generating capacity on Crabbs Peninsula, with a combined area of about 0.7 km$^2$ [41]. This is larger than the amount of space the CSP plants would occupy at around 0.4 km$^2$ for a 15 MW plant, which is the average plant size for the scenarios allowing for CSP. This is also comparable to the space required for a 40 MW plant, needing around 1.1 km$^2$, which is one of the largest plant sizes this study suggests.

The NDC target of installing solar panels on 30,000 houses [14], with an assumed 4 kW system for each house, requires around 2 km$^2$ of roofing space for a total of 120 MW of rooftop solar PVs. Since a typical house in Antigua and Barbuda has an area of 180 m$^2$ [41], the 30,000 homes represent approximately 5.5 km$^2$ of space, so more than half of all the roofing for these homes would be occupied with additional solar panels capable of being installed if the loads rise beyond the needs a 4 kW solar system. The entirety of most roofs in Antigua cannot be utilized for solar PVs, as the panels need to be flush to the roof, which requires more space, and the surface needs to be south-facing to obtain optimal output. Regardless, there should be sufficient space, with a margin of 3.5 km$^2$ of roofing, when compared to the 2 km$^2$ minimum calculated above. Parking areas and commercial, industrial, and government buildings would also represent surfaces that could be used for solar PVs without encroaching on new land. However, it does appear that additional land area on the order of 1.5–3.5 km$^2$ (on an island of 281 km$^2$) may be required for installing enough capacity to meet the target of 100% renewable energy.

### 3.9. Job Creation/Destruction

The estimates for the proposed systems found significant job creation in all cases. Figures 4 and 5 show job creation from 2020 to 2040 for a scenario with CSP. The almost 90 MW from the currently installed electricity system accounts for around 60 jobs based on the assumptions made to calculate the jobs required/created for this type of technology. The figures demonstrate that there is not only a short-term increase in job creation (mainly due to construction) but a long-term increase in operation and maintenance as well. Although the optimizations were run for a "green-field" assumption for 2035, in order to estimate employment, an S-curve interpolation plotted to that date was used as an approximation for the time dependence.

There are at least 500 jobs required for each of the scenarios that were analyzed. The technologies that create the most employment are solar and wind energy. Utility solar, especially, will require the greatest number of workers during construction because the size of the solar PV plants is substantially larger than any of the other technologies. As time goes on, it is clear that the number of jobs will continue to decrease after the peak in 2033, but this will still provide more jobs in Antigua and Barbuda than the current energy sector requires.

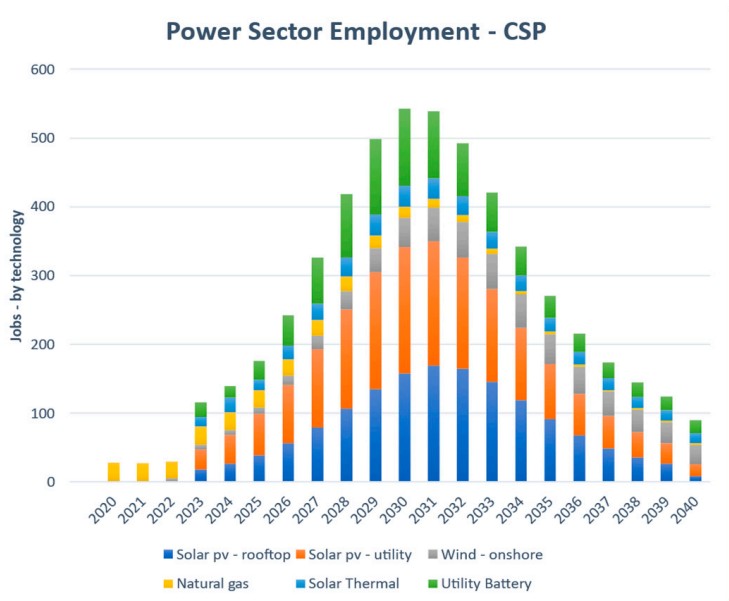

**Figure 4.** CSP scenario by technology.

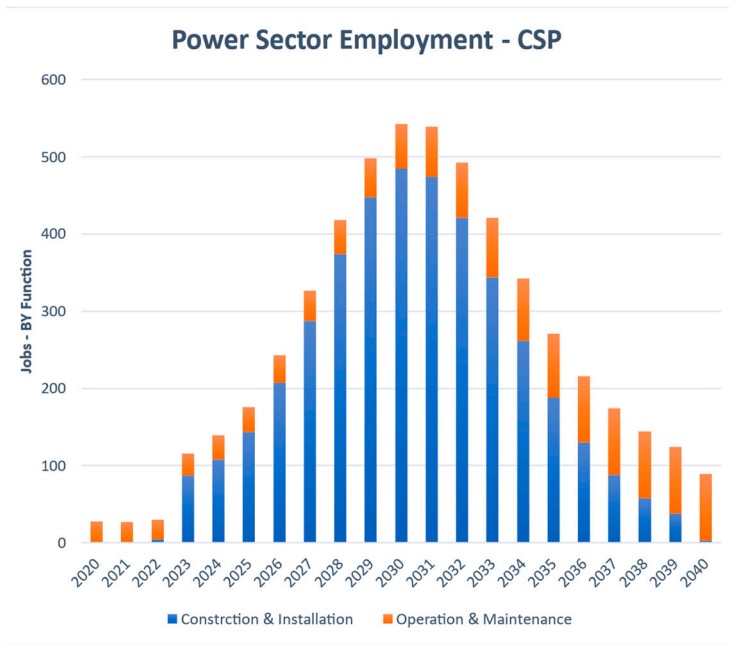

**Figure 5.** CSP scenario jobs by function.

## 4. Discussion

The results of this present study can be an important step toward understanding how more renewable energy can be implemented to meet Antigua and Barbuda's NDC target of 86% of energy being produced by renewable energy for 2030 and can be used to create a strategy to meet that target [14]. All the scenarios in our analysis achieve the target, and even in the scenario with remaining fossil-fuel technologies, diesel generation only accounts for 12% of total electricity generation. Thus, in an optimal system with the lowest LCOE of USD 83/MWh, the NDC target would already be met, with this being seen as a large step toward achieving a system with 100% renewable electricity. However, as shown in the sensitivity tests, the optimum scenarios found with remaining fossil fuel generation are sensitive to the assumption that fuel oil will remain inexpensive; the overall long-term cost of the system with higher fuel costs is equivalent to that without any diesel generation.

An advantage of the 100% renewable energy system is, therefore, the insensitivity to fluctuations in cost on global fossil-fuel markets.

The results in the present study are reflective of those found in IRENA's Renewable Energy Roadmap for Antigua and Barbuda [14]. That study analyzed the transition to a 100% Renewable Energy nation including the introduction of electric vehicles. Their results found LCOEs of around 0.10 USD/kWh (100 USD/MWh) for scenarios including small diesel plants and 100% renewable energy with hydrogen storage. The results found in this study are comparable to these results, with the addition of several more scenarios and a new technology, CSP, but with somewhat higher LCOEs in general.

The important goal of encouraging a "just transition", which is represented here by job impact considerations, can also be compared with a recent study. The results of the job impact assessment in this study can be compared to the results found in another recent study on Antigua and Barbuda's just transition [34]. The results found here show higher needs for jobs in construction and implementation but a smaller total of around 100 jobs longer-term jobs when compared to the 250 jobs found previously [34]. The energy system transformation clearly implies a transition in the nature of various economic sectors. To ensure a smooth and just transition to renewable energy, it is important to understand, in greater detail, the impact on the jobs and livelihoods of those working in traditional fossil fuel-based industries. This study shows that each scenario will require at least as many jobs as the current electricity sector has. As more projects are implemented, data gathering will be important to help policymakers develop effective strategies to support workers and mitigate any negative impacts of the transition to renewable energy.

One potential pathway forward from the current system would be to utilize the small-scale diesel generation that many businesses and homes already have for emergency generation when the grid is unable to provide adequate energy across the island. These types of generators are able to power on and off easily and are more flexible than the large-scale 10 MW generators. Thus, in the scenarios where the diesel generators are limited to 10 and 5 MW and where the LCOE is lower than in any other scenario, it may be feasible to rely on these small-scale generators for distributed backup generation for those times of the year that the renewables and storage cannot meet. Figures 2 and 3 in the results section show how many of these hours are concentrated between the months of November to February.

Since many households and businesses already own diesel backup generators, a system solution could be feasible if co-ordination between distributed generation and the grid could be managed. Such a coordinated distributed system may also be a step towards 100% renewable energy generation, in which household solar generation, batteries, and electric vehicles will provide the backbone for a combined smart grid. One important issue that will have to be considered is that of the distribution of the burden of costs for a transition; if, as suggested here, the renewable energy solution is the most cost-effective one, at least over the lifetime of the system, provisions will have to be made to ensure that those with lesser financial means can be part of that transition and not have to rely on the more expensive fossil-fuel options, such as personal backup generators.

In one of the main contributions of the present work, we find that the implementation of CSP helps to significantly drop the LCOE of the system and allows for storage times that bridge the relatively rare longer periods of low solar PV and wind energy, compared to those scenarios with only variable renewables and utility-scale batteries. For most scenarios and under most constraints, the optimization model chose to include CSP as part of the system, showing that the technology is well-suited for the island context. These results can be extrapolated to model energy systems for other nations in the Caribbean or island states in other regions around the world. The capability to expand this study to other nations and regions would be valuable when considering Antigua has a bare repertoire of renewable energy resources with no dispatchable renewable energy sources.

Another key outcome of the scenario selection is that increasing wind power from 25 to 50 MW decreases the LCOE of the system noticeably for every combination of

technologies, but the LCOE will only decrease very slightly with systems larger than 50 MW. Finally, the implementation of hydrogen generation and storage will create a similar reduction in LCOE as CSP. Our results provide a range of possibilities, thus allowing policymakers to gain a better understanding of how different technologies might perform in different contexts, which can inform decisions about where to invest resources and which technologies to prioritize.

Another important consideration is public and governmental trust in different technologies. In some scenarios, certain technologies may be more widely accepted and trusted, while others may face more resistance or skepticism. By considering scenarios that reflect varying levels of public trust, policymakers can identify the potential barriers to the adoption of certain technologies and develop strategies to address them. This could involve investing in public education campaigns or conducting outreach to build trust and understanding around specific technologies.

In conclusion, these results show that there is no single defined pathway towards a 100% renewable energy system. These results are likely reflected in other nations in the Caribbean that share similar resources and current energy sectors. However, in most other Eastern Caribbean countries, there are geothermal or hydropower resources that can be effectively used as dispatchable resources to enable greater penetration in terms of the cheapest renewable energy sources: wind and solar PVs. The present work shows that even in the more challenging case of Antigua and Barbuda, 100% renewable electricity systems are viable and are significantly less costly than current power systems.

**Author Contributions:** Conceptualization, R.J.B.; methodology, P.H., A.C. and R.J.B.; validation, A.C. and R.J.B.; investigation, P.H.; data curation, R.J.B.; writing—original draft preparation, P.H. and R.J.B.; writing—review and editing, P.H., A.C. and R.J.B.; visualization, P.H. and R.J.B.; supervision, A.C. and R.J.B. All authors have read and agreed to the published version of the manuscript.

**Funding:** This research received no external funding. PH acknowledges support from the University of Dayton Honors Program for his Honors Thesis work.

**Data Availability Statement:** Data files, CSP and PV model spreadsheets, PyPSA code, and Jupyter notebooks are available on GitHub at https://github.com/RJBrecha/Hoody_Antigua (accessed on 22 August 2023).

**Acknowledgments:** We acknowledge support from Jonas Hörsch from Climate Analytics in Berlin, Germany, for useful conversations concerning the implementation of PyPSA.

**Conflicts of Interest:** The authors declare no conflict of interest. R.J.B. has worked in the past with Climate Analytics and the Antigua and Barbuda Department of Environment colleagues on energy models used for developing nationally determined contribution (NDC) targets; however, the present work is independent of those efforts and does not imply the approval of the Department of Environment in Antigua and Barbuda.

## Appendix A. Levelized Cost of Electricity

The levelized cost of electricity (LCOE) is an important characteristic of an energy system to determine if it will be an economically viable system for a given load. The LCOE is a value that is found by adding up the lifetime discounted cost of an electricity plant, including the capital and marginal costs, divided by the total units of electricity the plant will generate in its lifetime. In the models used in this paper, the LCOE for the system is given in terms of USD/MWh of electricity generated by the system as a whole, not for individual technologies. All of the capital and marginal costs associated with a given system in this analysis can be added together and then divided by the total yearly load to find the LCOE for the given system. Thus, the lower the LCOE that is calculated, the more economically viable the system will be.

### Appendix B. Capital Recovery Factor Scaling

The capital recovery factor is the ratio of a constant annuity to the present value of receiving that annuity for a given length of time [42]. This capital recovery factor can be defined using the following equation:

$$\text{CRF} = \frac{i * (i+1)^n}{(i+1)^n - 1} \tag{A1}$$

The variable $i$ is the discount rate, and the variable $n$ is the lifetime of the plant in years. In this scenario, $i$ was assumed at a value of 7%, or 0.07, and $n$ was assumed at a conservative value of 25 years (15 years was used for utility batteries). When using Equation (A1) and multiplying the values in Tables 1 and 2 from the methodology section, this results in the values shown in Tables A1 and A2 below.

**Table A1.** CRF multiplied by the generator and link capital costs, together with marginal costs.

| Generator | Capital Cost (USD/kW) | Marginal Cost (USD/MWh) |
|---|---|---|
| Diesel | 154.46 | 170 |
| Wind | 115.84 | N/A |
| Solar PV | 75.51 | N/A |
| Concentrating Solar Power Solar Field | 226.54 | N/A |
| Concentrated Solar Power Turbine | 65.22 | N/A |
| Hydrogen Electrolyzer | 85.81 | N/A |
| Hydrogen Fuel Cell | 42.91 | N/A |

**Table A2.** CRF multiplied store capital costs.

| Stores | Capital Cost (USD/kWh) |
|---|---|
| Utility Battery | 143 |
| CSP Thermal Storage | 50 |
| Hydrogen Storage | 33.33 |

### Appendix C. Employment Factors and Job Creation

The values for the total capacity of each technology and the capacity added each year are used in conjunction with employment factors to find jobs/MW and job years/MW for each technology in terms of construction and installation (C&I) and operation and maintenance (O&M), as well as the total job impact of the jobs in the electricity sector [35]. This was carried out on a yearly basis using a logistic curve to implement the system over a given period of time, as this would be more realistic than implementation conducted all at once. Then, to find the required diesel system, an approach that found the kWh/kW power for renewable energy generator technology was used, and then we multiplied the found MW capacity calculated from the logistic curve of these technologies. These total generation values were summed together and subtracted from the yearly load and then divided by the number of hours in a year: 8760 h. Finally, this was adjusted by multiplying the value by 1.9 because the current system is much larger than just the load divided by the number of hours due to maintenance, large peak loads, etc.

Job creation analyses were conducted for several scenarios, including those without CSP and with hydrogen storage and without hydrogen or CSP. All those scenarios show that solar PV will dominate job creation but not necessarily job sustenance. Utility battery storage appears to have a significant portion of job sustenance, but overall, the technologies share similar portions of the job needs of the entire electricity sector. Wind energy will be another large contributor to job creation for those scenarios. Additionally, all the scenarios follow a similar trend in showing a large increase in jobs until 2033, where it will peak and then slowly but slightly decrease and plateau.

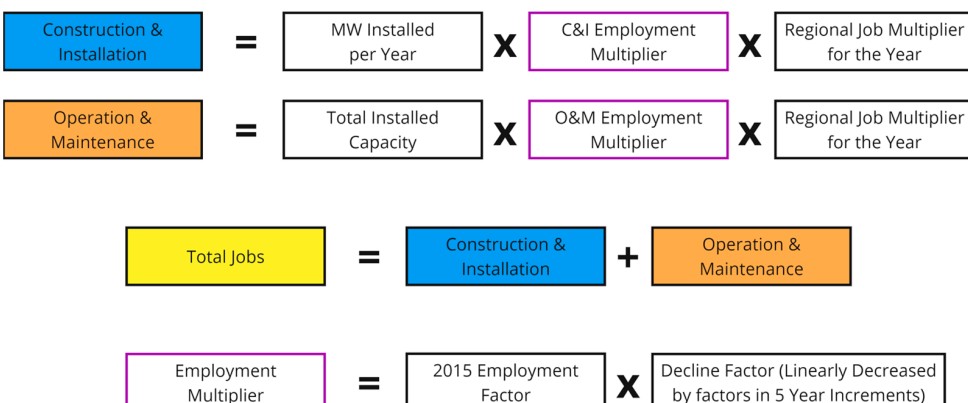

**Figure A1.** Job impact considerations.

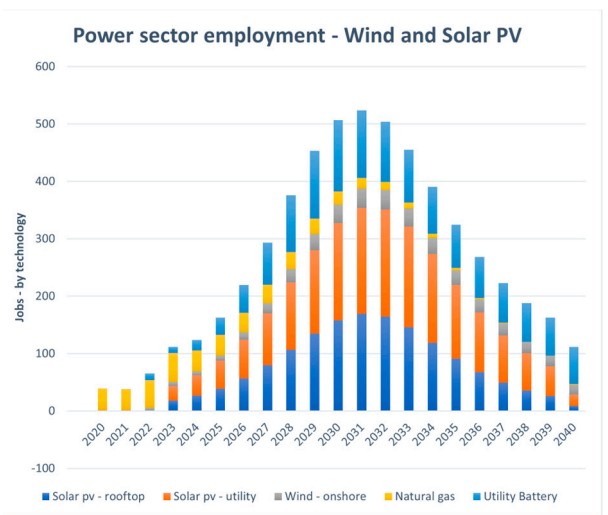

**Figure A2.** Wind and solar PV scenario jobs by technology.

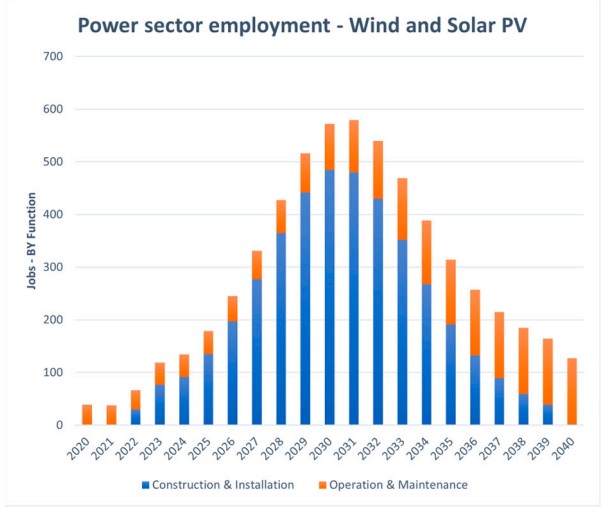

**Figure A3.** Wind and solar PV scenario jobs by function.

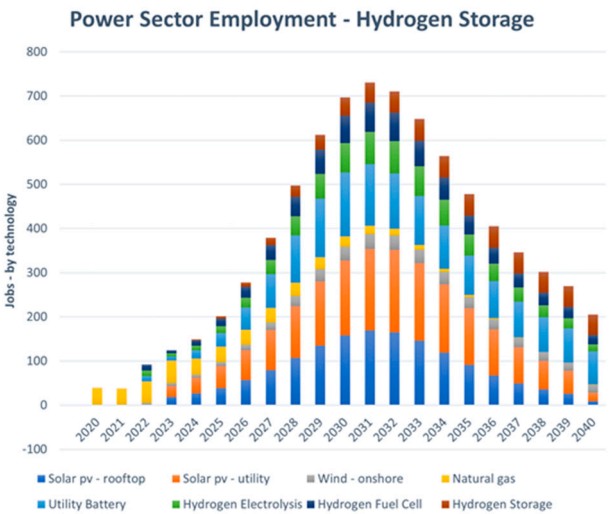

**Figure A4.** Hydrogen scenario jobs by technology.

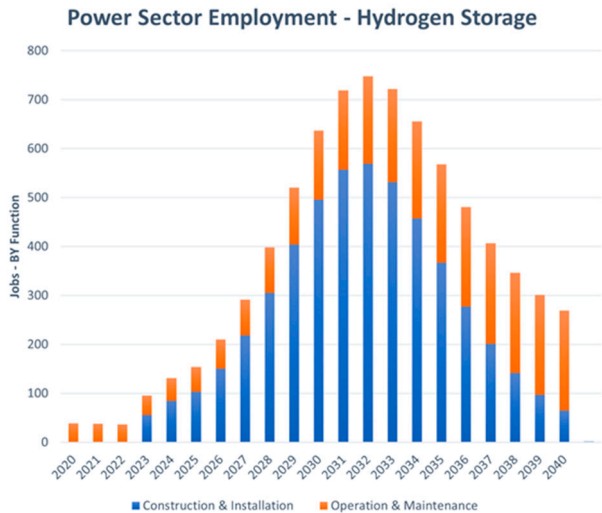

**Figure A5.** Hydrogen scenario jobs by function.

### Appendix D. Details of CSP Calculations

The PyPSA model requires hourly input data, but since typical, representative hourly solar radiation data are not readily available for the project site, we used satellite-derived data for the year 2021 from NASA POWER [24]. The hourly data included the clearness index, air temperature, latitude, longitude, and horizontal solar radiation. The monthly average daily solar radiation for the year 2021 from NASA POWER [24] is compared to the typical values from NASA POWER [24] in Figure A6. An inspection of Figure D-1 reveals that the average daily solar radiation for 2021 varies little from the typical values, with differences ranging from 0.4% in January to about 10% in November. On an annual basis, the average daily solar radiation in 2021 was 2.75% higher than typical.

The projected hourly output of the electricity of the PV arrays and CSP plant were used as the input for the PyPSA model. To utilize the total horizontal solar irradiation data from NASA, a methodology was needed to estimate the beam and diffuse fractions, and thus, the algorithms described in [43] were used in this study. To summarize, Equations (A2) and (A3) are first used to find the diffuse radiation on the tilted surface. Equation (A4) is used to find the solar radiation on the tilted surface; Equation (A5) calcu-

lates the ground diffuse radiation that is combined with the sky diffuse radiation for solar PV. The process is shown in Figure A7.

$$I_d = I_{dH} \tag{A2}$$

$$I_{d\theta} = I_d \times F_{SS} \tag{A3}$$

$$I_{D\theta} = \left( \frac{I_{DH}}{\cos \theta_H} \right) \times \cos \theta \tag{A4}$$

$$I_{R\theta} = I_{TH} \times \rho \times F_{SG} \tag{A5}$$

where $I$ is the hourly solar radiation, $F_{SS}$ is the surface-sky view factor, and $\theta$ is the solar incidence angle; subscript $d$ is diffuse; subscript $D$ is beam.

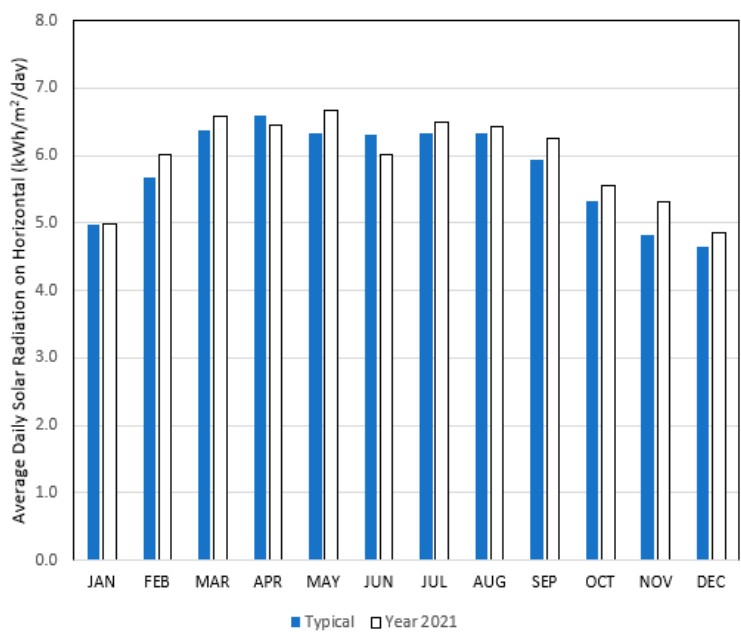

**Figure A6.** Comparison of typical monthly average daily (horizontal) solar radiation to the actual values from 2021.

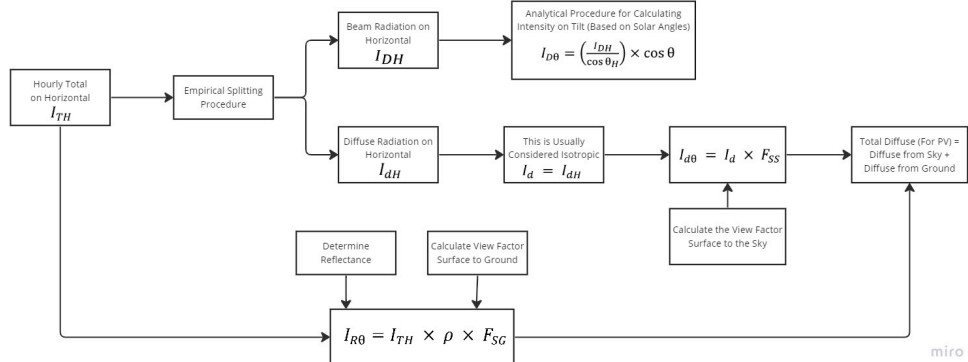

**Figure A7.** Calculating beam radiation from total horizontal.

The useful heat output from the concentrating parabolic array was calculated from the beam radiation, ambient temperature, the fluid temperature entering the collectors, array size, and the solar incidence angle using algorithms from Duffie and Beckman [43]. The useful thermal energy from the solar array was then used to calculate the amount of energy

converted to electrical energy by calculating the actual Rankine power cycle efficiency as a function of Carnot efficiency. The total solar radiation on the tilted PV surface, in conjunction with the rated efficiency corrected for the current hourly PV cell temperature, was used to determine the PV electrical energy output. The key differences between the solar PV and CSP calculations are that the solar PV uses total diffuse (ground and direct), whereas CSP uses direct radiation, and the CSP model has the capability to generate up to its nameplate capacity; PV only reaches about 75% of its nameplate capacity due to inherent losses in conversion from DC to AC power.

Data files and the CSP model spreadsheets are available on GitHub at https://github.com/RJBrecha/Hoody_Antigua (accessed on 22 August 2023). PyPSA only needs a scaled output for relative capacity, with the optimization "finding" the optimal capacity and the time-dependent output. The storage time, or more precisely, the ratio of energy storage capacity (MWh) to power output (MW) is maximized at 24 h, which the modeling determined as resulting in the lowest LCOE.

As mentioned in the Methodology, the capital costs of the CSP generation and storage are described based on the electrical output and storage size. The storage capital costs were found using the proportions of real CSP plant cost breakdowns. Based on studies such as two of IRENA's studies, a breakdown of the total capital cost of 85% for generation and 15% for storage was used to find the cost per MWh [31,44]. This is reflected in the GitHub material.

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
