# Peer review of "The Transition to a Renewable Energy Electric Grid in the Caribbean Island Nation of Antigua and Barbuda"

_energies, doi:10.3390/en16176206_

Round 1

Reviewer 1 Report

Dear authors:

The article is very interesting and current, the ambitious process of generating 100% of energy from renewable sources is a global goal, especially for islands with so few resources and high energy dependence. Before publication, some aspects should be improved:

-The objectives and contributions to the scientific community should be defined in the abstract, introduction and linked in the conclusions.

They need to compare your results with other studies, it is important to make an exhaustive state of the art.

-Methodology:

How does the model validate wind and solar potential?

Which studies are the CAPEX and LCOE results based on?

How can the study be extrapolated to another island?

In general they should detail the methodology so that it can be applied to another region.

Regards:

Reviewer 2 Report

The manuscript lacks details concerning the discussion of the meteorolodical an operation conditions of a CSP system for th site speifcic meteorological conditions.

The storage sizing  shoud be discussed explicitly in view of the differences in the site specific time series chararacteristics of the irradiance data

language quality is ok

Round 2

Reviewer 1 Report

Thanks for the modifications and explanations made, Figure 1 brings a lot of quality to the work.

Author Response

Thank for your comment.

Reviewer 2 Report

Manuscript is still missing adequate information on the schene used to calculate the time series of the CSP power output, including information on its accuracy. Information on the solar (PV and CSP) generation data is partly confusing " An hourly output of each generator was needed for PyPSA modeling. The solar PV 189 and solar CSP used hourly output for the systems using the NASA POWER database 190 which has hourly data for various weather conditions from several years for Antigua [24]. " Reference 24 is: "M. Wilson, “Lazard’s Levelized Cost of Storage Analysis—Version 4.0,” LAZARD, Nov. 2019 " which does no realy fit here.

Information on the accuracy of  time series characteristics, essential for storage sizing is not given.
